# An enhanced correlation identification algorithm and its application on spread spectrum induced polarization data

Siming He[1,2], Jian Guan[3], Xiu Ji[1,4], Hang Xu[1], and Yi Wang[2]

[1]School of Electrical and Information Engineering, Changchun Institute of Technology, Changchun 130000, China
[2]College of Instrumentation and Electrical Engineering, Jilin University, Changchun 130000, China
[3]College of Electronic Science & Engineering, Jilin University, Changchun 130000, China
[4]National Local Joint Engineering Research Center for Smart Distribution Grid Measurement and Control with Safety Operation Technology, Changchun Institute of Technology, Changchun 130000, China
*Correspondence 1 to:* Yi Wang (e-mail: wangyijlu@jlu.edu.cn)

*Correspondence 2 to:* Xiu Ji (e-mail: jixiu523@163.com)

**Abstract.** In spread spectrum induced polarization (SSIP) data processing, attenuation of background noise from the observed data is the essential step that improves the signal-to-noise ratio (SNR) of SSIP data. The Time-Domain Spectral Induced Polarization Based on Pseudo-random Sequence (TSIP) algorithm has been proposed to improve the SNR of these data. However, signal processing in background noise is still a challenging problem. We propose an enhanced correlation
identification (ECI) algorithm to attenuate the background noise. In this algorithm, the cross-correlation matching method is helpful for the extraction of useful components of the raw SSIP data and suppression of background noise. Then the frequency-domain IP (FDIP) method is used for extracting the frequency response of the observation system. Experiments on both synthetic and real SSIP data show that the ECI algorithm can not only suppress the background noise but also better preserves the valid information of the raw SSIP data to display the actual location and shape of adjacent high resistivity anomalies, which
can improve subsequent steps in SSIP data processing and imaging.

## 1 Introduction

Induced Polarization (IP) technology operated in both the time domain and the frequency domain is useful in exploration for groundwater mapping, mineral exploration, and other environmental studies (Revil 2012, 2019; Høyer et al. 2018). Since the phenomenon of IP in time domain was first discovered by Schlumberger in 1920s, there has been consistent efforts to explore
its utilization in various researches. In 1959, the frequency-domain IP (FDIP) approach is proposed by Collett and Seigel, which became the most classic and widely used mapping technique. For example, the first variable-frequency approach was proposed by Wait et al in 1959, then the spectrum approach of the complex resistivity was developed by Zonge and Wynn in 1975, and the dual-frequency IP approach was presented and developed by He et al. in 1993 and Han et al. in 2013. Recently, spread spectrum induced polarization (SSIP) is a popular branch of FDIP which uses pseudo-random current pulses of opposite
polarity as an excitation source (Chen et al., 2007; Xi et al., 2013, 2014; He et al. 2015). According to the intrinsic broadband characteristics of the source itself, the spectral response of an observation system can be simultaneously calculated at multiple frequencies in electrical exploration (Liu et al., 2019). Thus, this SSIP technology has been gaining attention and application in electrical prospecting (Xi et al., 2014; Lu et al., 2019; Wang and He, 2020).

In field detection experiments, it is still a major problem that IP data is often contaminated with background noise. The
35 background noise can be mainly categorized into two types: the Gaussian noise and the impulsive interference with different percentage of outliers (Liu et al., 2016; Kimiaefar et al., 2018; Li et al., 2019). If the background noise is not effectively reduced, the remnant noise can affect the calculation of complex resistivity and may mislead subsequent interpretations of the subsurface structure.

The field of FDIP denoising has achieved quite good results through the constant research of experts and scholars. There
have been many algorithms that can be used to suppress the FDIP random noise (Mo et al., 2017), such as smooth filter (Guo,

2017), Mean stack (Liu, 2015), digital filter (Meng et al., 2015), and robust stacking (Liu et al., 2016). The smooth can effectively attenuate Gaussian noise, but the impulsive interference with intense energy leaves the effectiveness of this algorithm limited. Therefore, an effective attenuative algorithm for background noise is still a challenging task for traditional noise suppression algorithms (Neelamani et al., 2008; Liu et al., 2017). SSIP method also faces the same issue (Liu et al., 2016, 2017).

Recently, the new algorithm based on a circular cross-correlation method, Time-Domain Spectral Induced Polarization Based on Pseudo-random Sequence (TSIP) algorithm, has also been used to suppress the SSIP noise (Li et al. 2013; Zhang et al. 2020). Due to its effective denoising ability, the identification method has gained more attention and development. However, the TSIP algorithm is restricted because the excitation signal is sensitive to the random noise. For this problem, we propose an enhanced correlation identification (ECI) algorithm for reducing the noise in SSIP data. The ECI algorithm obtains cross-correlations between the transmitter output signal, the excitation signal, and the response signal. The performance of the ECI algorithm is demonstrated on both synthetic and field SSIP data. Experimental results show that the ECI algorithm can effectively control the root mean square of noise (NRMS) increase, enhance its denoising performance in background noise and improve the valid signal preservation to display the actual location and shape of high resistivity anomalies with higher resolution.

## 2 Theory

### 2.1 Subsection (as Heading 2) The TSIP theoretical model

Figure 1 shows a traditional diagram of the electrical resistivity survey. The transmitter output signal $u_{\mathrm{T}}(t)$ is poured from electrode A to electrode B, the excitation signal $i(t)$ flows from electrode A to electrode B, and the response signal $u(t)$ between the electrodes M and N is measured. To simultaneously obtain the spectral response of subsurface at various frequencies, pseudo-random sequence based the excitation signal $i(t)$ is considered. Thus, the spectral response of subsurface be retrieved by the TSIP algorithm, and its spectral response be expressed as (Li et al., 2013):

$$H_e(\omega) = \frac{P_{ui}(\omega)}{P_{ii}(\omega) \cdot P_S(\omega)}, \tag{1}$$

where $P_{ui}(\omega)$ is the cross-power spectral density of $u(t)$ and $i(t)$, $P_{ii}(\omega)$ the auto-power spectral density of $i(t)$, and $H_S(\omega)$ is the impulse spectral response of the observing system.

Given this observation mode using low-power signals, the magnetotelluric system is a time-invariant system and let us suppose that $H_S(\omega)$ is 1. Eq. (1) can further be expressed as:

$$H_e(\omega) = \frac{P_{ui}(\omega)}{P_{ii}(\omega)} = \frac{fft[R_{ui}(\tau)]}{fft[R_{ii}(\tau)]} = \frac{U(\omega)}{I(\omega)}, \tag{2}$$

where $fft[.]$ denotes Fast Fourier Transform (FFT), $R_{ui}(\tau)$ is the cross-correlation function of $u(t)$ and $i(t)$, $R_{ii}(\tau)$ is the auto-correlation function of $i(t)$, $U(\omega)$ and $I(\omega)$ depict the geometric factor defined by the frequency spectrum of $u(t)$ and the frequency spectrum of $i(t)$ respectively, and $\tau$ denotes time-delay.

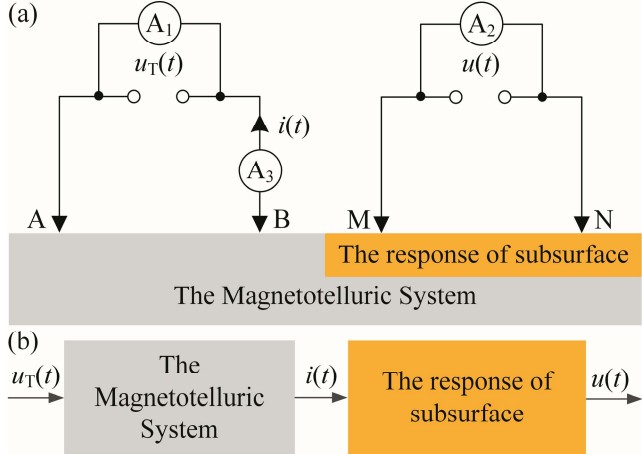

**Figure 1.** (a) The observation model of the four-electrode measurement. (b) its equivalent diagram.

In the practical field environment, this observation mode is contaminated by the background noise, as shown in Figure2. The output of the sensors $A_k$ ( $k=1,2,3$ ) can be expressed as:

$$y_1 = u_T(t) + n_1(t), \tag{3}$$

$$y_2 = u(t) + n_2(t), \tag{4}$$

$$y_3 = i(t) + n_3(t), \tag{5}$$

where $n_k(t)$ is the background noise.

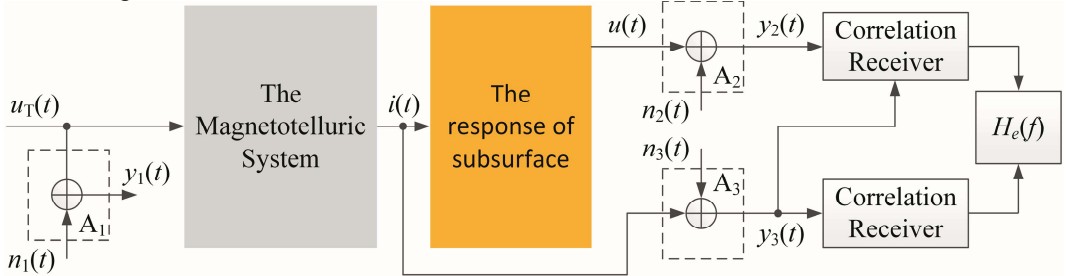

**Figure 2.** Schematic diagram using the TSIPD algorithm.

Therefore, according to Eq. (2), the formula of the TSIP algorithm is given as:

$$H_e(\omega) = \frac{P_{y_2 y_3}(\omega)}{P_{y_3 y_3}(\omega)} = \frac{fft[R_{y_2 y_3}(\tau)]}{fft[R_{y_3 y_3}(\tau)]} = \frac{fft[R_{ui}(\tau) + R_{un_2}(\tau) + R_{in_1}(\tau)]}{fft[R_{ii}(\tau) + R_{in_1}(\tau) + R_{n_1 n_1}(\tau)]} \approx \frac{fft[R_{ui}(\tau)]}{fft[R_{ii}(\tau) + R_{n_3 n_3}(\tau)]}. \tag{6}$$

Eq. (6) demonstrates that the TSIP algorithm has a weak denoising effect when $n_3(t)$ is the massive intense noise. In other words, the TSIP algorithm depends on the energy intensity of $n_3(t)$ present in $i(t)$ .

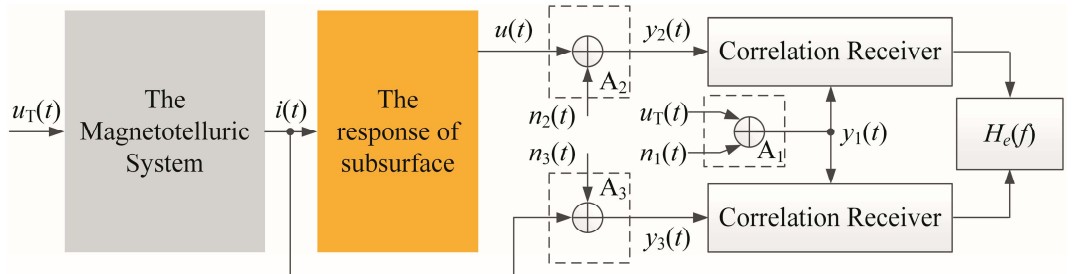

**Figure 3.** The schematic diagram of the ECI denoising model.

**2.2 The ECI theoretical model**

That the denoising ability of the TSIP algorithm is limited is caused by that $i(t)$ is sensitive to $n_3(t)$. To solve this problem, the ECI algorithm is proposed and its derivation process is as follows.

     Firstly, let us suppose that the telluric system is a time-invariant system under low-power signals. For three sensor output signals, their cross-correlation functions are the periodic correlation functions of time $\tau$. When the length of the correlation window $NT$ is specified, 0.0125s in this experiment. The cross-correlation functions can be expressed as follows:

$$R_{y_1y_2}(\tau)=\mathrm{E}[y_1(t)y_2(t\text{-}\tau)]=(R_{u_Tu}(\tau))_N + R_{n_1n_2}(\tau), \tag{7}$$

$$R_{y_1y_3}(\tau)=\mathrm{E}[y_1(t)y_3(t\text{-}\tau)]=(R_{u_Ti}(\tau))_N + R_{n_1n_3}(\tau), \tag{8}$$

where $R_{n_1n_2}(\tau)$ and $R_{n_1n_3}(\tau)$ are the cross-correlations of, $n_2(t)$ and $n_3(t)$ respectively, and $\tau$ is time-delay that lies in the range of $-NT$ to $NT$.

     Figure 4 shows the schematic diagram of ZW-CMDSII (Zhang et al., 2014; He et al., 2014;). As is known from the figure, we can conclude that $u_T(t)$ is mainly disturbed by the floor noise energy of the instrument, and $i(t)$ and $u(t)$ are mainly 15   contaminated by environmental noise. The floor noise is relatively very low, while environmental noise possesses a much higher energy level. Thus we assume that $n_1(t) \approx 0$, and can conclude that zero correlation between $n_1(t)$ and $n_2(t)$, $n_3(t)$, $R_{n_1n_2}(\tau) \approx 0$ and $R_{n_1n_3}(\tau) \approx 0$.

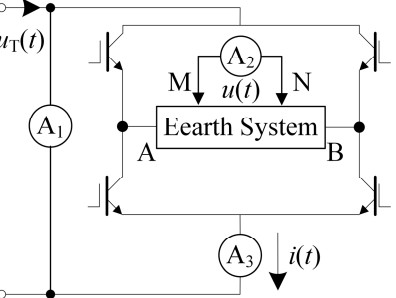

**Figure 4.** Schematic diagram of the instrument.

Based on the above analyses, we can further obtain:

$$R_{y_1y_2}(\tau) \approx (R_{u_Tu}(\tau))_N, \tag{9}$$

$$R_{y_1y_3}(\tau) \approx (R_{u_Ti}(\tau))_N. \tag{10}$$

Then the cross-power spectrum of Eq. (9) and Eq. (10) can be written as following

$$P_{y_1y_2}(\omega) \approx P_{u_T u}(\omega), \tag{11}$$

$$P_{y_1y_3}(\omega) \approx P_{u_T i}(\omega). \tag{12}$$

Finally, according to Eq. (2) and Eq. (11), Eq. (12) can be expressed as following

$$5 \quad H_e(\omega) = \frac{U(\omega)}{I(\omega)} = \frac{U(\omega)U_T^*(\omega)}{I(\omega)U_T^*(\omega)} = \frac{P_{u_T u}(\omega)}{P_{u_T i}(\omega)} \approx \frac{P_{y_1y_2}(\omega)}{P_{y_1y_3}(\omega)} = \left| \frac{P_{y_1y_2}(\omega)}{P_{y_1y_3}(\omega)} \right| e^{-j(\varphi_{y1y2}(\omega) - \varphi_{y1y3}(\omega))} \tag{13}$$

where $\varphi_{y1y2}(\omega)$ and $\varphi_{y1y3}(\omega)$ denotes the difference between $y_1(t)$, $y_2(t)$ and $y_3(t)$.

So, Eq. (13) is the formula of the ECI algorithm. The derivation process of this formula clearly describes that the ECI algorithm can effectively suppress the background noise and be independent on the degree of $n_3(t)$ present in $i(t)$.

### 3 Experiment on synthetic SSIP data record

10  We test the ECI algorithm for attenuating background noise of SSIP data sets in comparison with the FDIP algorithm and the TSIP algorithm. For the comparison, the SNR, root mean square of noise (NRMS) and relative error ($\varepsilon$) as the objective parameters to judge the performance of denoising, which are calculated as follows:

$$\text{SNR} = 10\log_{10}\left\{ \frac{\sum_{i=1}^{M}[y(i) - \mu_y]^2}{\sum_{i=1}^{M}[n(i) - \mu_n]^2} \right\}, \tag{14}$$

$$\text{NRMS} = \sqrt{\frac{\sum_{i=1}^{M}[n(i)]^2}{M}}, \tag{15}$$

$$15 \quad \varepsilon = 100 \times \frac{\rho_1 - \rho_0}{\rho_0}, \tag{16}$$

$$i(t) = u_i(t) / R_S \tag{17}$$

where $\mu_y$ and $\mu_n$ denote the mean values of the useful signal and the noise separately. $y(i)$ and $n(i)$ are the useful signal and the noise separately, M is the length, $\rho_0$ denotes the complex resistivity calculated without noise, and $\rho_1$ is the complex resistivity calculated with the noise added to $\rho_0$. $R_S$ is the value of the sampling resistor ($R_S = 1\Omega$), and $u_i(t)$ is the voltage at the sampling resistor.

20   To validate the effectiveness of the ECI system, we performed a resistance-capacitance experiment, as shown in Figure 5. The circuit parameters are chosen to be $R_A = 30.3\Omega / 5W$, $R_{MN} = 30.1\Omega / 5W$, $R_B = 30\Omega / 5W$ and $C_{MN} = 470\mu F$. We recorded the applied voltage $u_T(t)$, the injected current $i(t)$ and the measured potential signal $u(t)$ as the raw signals. These signals are a 3-order spread spectrum pseudo-random sequence at the clock cycle of 0.0125s, as shown in Figures 6a-6c and Table 1.

Since our experiment is in a stable environment, we consider the system linear time-invariant and the noise from the current and voltage measurement are linearly superpositioned (Pelton, et al., 1983; De, et al., 1983; Vinegar and Waxman, 1984; De, et al., 1992; Garrouch

and Sharma, 1998). Therefore, it is actually equivalent whether the noise is added to the injected current $i(t)$, the measured potential signal $u(t)$ or the applied voltage $u_T(t)$. Therefore, the injected current $i(t)$ is only polluted by the synthetic background noise, including Gaussian and impulsive, as shown in Figures 6d and 6e. Thirdly, the complex resistivity of the main frequency is considered and discussed because the main energy of the pseudo-random signal is concentrated on the main frequency (He, 2017). Finally, for detailed comparisons between the ECI algorithm and the others, we add the synthetic Gaussian and impulsive noises to the response signal $i(t)$, respectively.

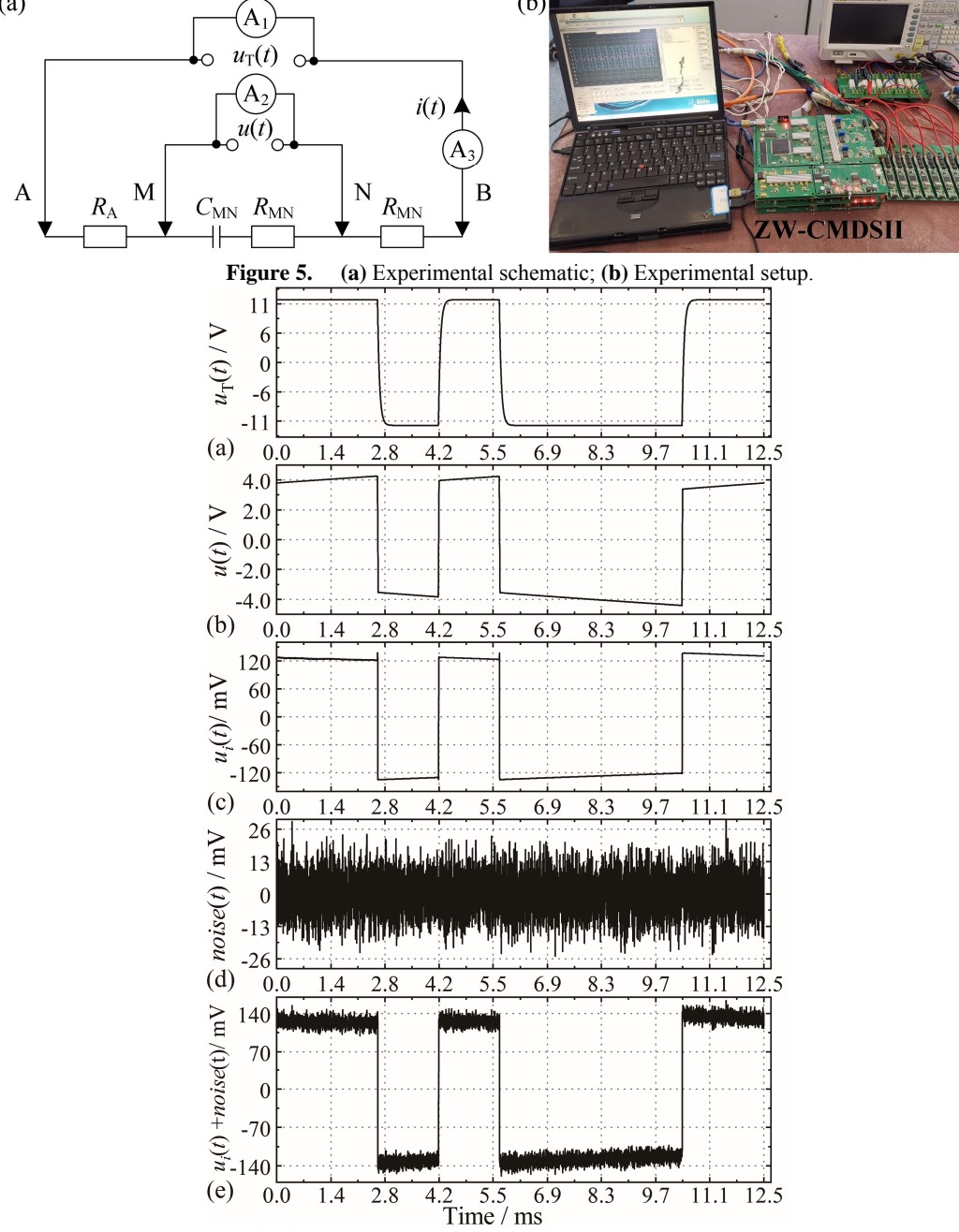

**Figure 5.** (a) Experimental schematic; (b) Experimental setup.

**Figure 6.**    The time waves of **(a)** the applied voltage $u_T(t)$, **(b)** the measured potential signal $u(t)$, **(c)** the voltage $u_i(t)$ at the sampling resistor, **(d)** *noise*(*t*), and **(e)** the synthetic signal ($u_i(t)$+ *noise*(*t*)).

**Table 1.**    Amplitude and phase values of complex resistivity obtained with Figures 6a-6c.

| Frequency (Hz) | Theoretical amplitude (Ω) | Theoretical phase (rad) | Measured amplitude (Ω) | Measured phase (rad) |
|---|---|---|---|---|
| 80.2 | 30.8 | -0.14 | 30.8 | -0.14 |
| 160.4 | 30.4 | -0.07 | 30.3 | -0.08 |
| 320.8 | 30.2 | -0.03 | 30.7 | -0.03 |

   We use synthetic Gaussian noise with the deviation and mean values of 0.1 and 1.1 as a standard template. The excitation signal $i(t)$ is polluted by synthetic different energy levels of the Gaussian noise. Figure 7 shows that the denoised results are obtained and compared at the three main frequencies when the noise RMS ranges from 0.12 to 0.25. The figure shows that as the RMS of noise increases, the complex resistivity information obtained by each algorithm decreases. However, the amplitude spectrum after ECI processing has the slowest falling speed, and the phase spectrum has the slowest falling speed at 80 Hz.

**Figure 7.**    Amplitude and phase of complex resistivity values at **(a1)** and **(b1)** 80 Hz, **(a2)** and **(b2)** 160 Hz, **(a3)** and **(b3)** 320 Hz comparison using the three methods.

   Previous literature has shown that if the percentages of outliers in impulsive noise exceed $50\%$, the traditional denoising algorithm will be limited (Liu et al., 2016, 2017). Thus, Synthetic impulsive noise is added to the excitation signal $i(t)$ in ten percent steps. Their standard deviations (SDs) and skewnesses (SKs) are shown in Figure 8. As depicted in Figure 9, the three algorithms have a certain degree of denoising performance versus the different percentages of the synthetic outliers against the raw data. The figure shows that with the discrete points of impulse noise growing, the RMS of noise is different. The amplitude spectrum and phase spectrum of complex resistivity obtained by each algorithm fluctuate. The amplitude spectrum after ECI processing remained the slowest falling speed. Although the noise reduction performance of the phase spectrum processed by ECI does not stand out, the overall change of the amplitude spectrum after ECI processing is still slow, especially when the discrete point is more than 60%.

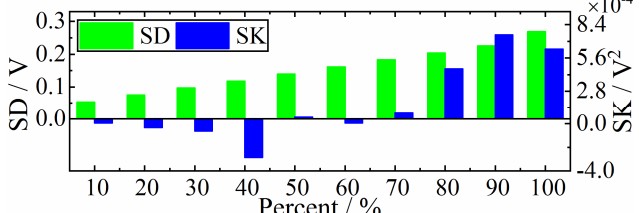

**Figure 8.** The standard deviations (SDs) and skewnesses (SKs) of synthetic impulsive noise.

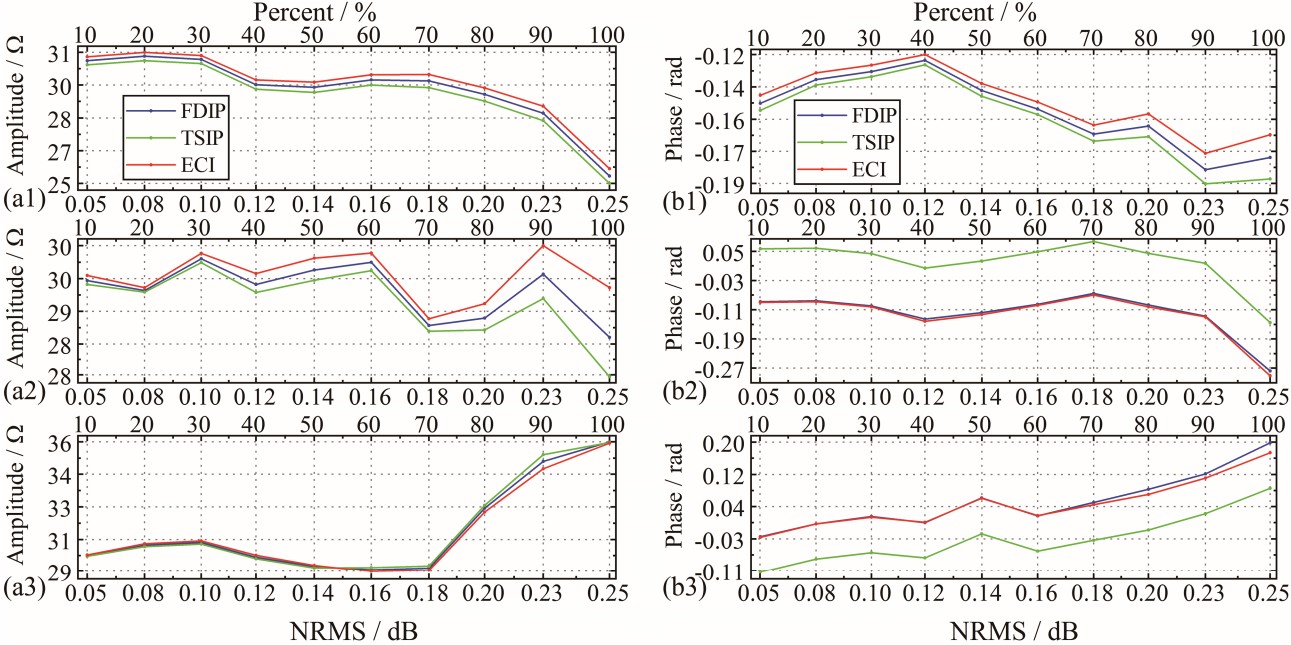

**Figure 9.** Complex resistivity values at **(a1)** and **(b1)** 80 Hz, **(a2)** and **(b2)** 160 Hz, **(a3)** and **(b3)** 320 Hz comparison using the three methods.

## 4 Experiment on real SSIP data record

To further verify the performance of the ECI algorithm, the Wenner array, the traditionally applied system in the field, was selected for performing laboratory tests, as shown in Figures 10 and 11. SSIP data was acquired with high-density meter and 20 electrodes at 1m spacing. A Wenner acquisition sequence was adopted with 55 potential measurements expressed utilizing the green and points. The figure shows an example of two high resistance cavities. The two cavities were presented by the letters A and B, and their calibers were about 1.8 m × 2 m. The two cavities are buried by loess. The loess is measured to have an electronic resistivity of $36 \, \Omega \cdot m$. The measured excitation signal had a range between 0.04 and 0.19 A approximately. The transmitter output signal is a three-order sequence with 80 Hz frequency, and its voltage is about ± 11.8 V. The sampling frequency is 625 kHz. The excitation and response data of 40 periods were recorded at each point.

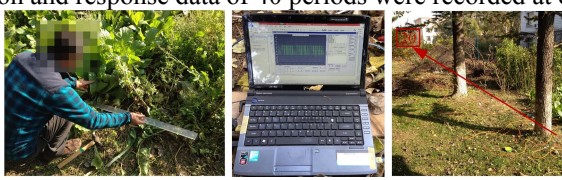

**Figure 10.** Diagram of the field-test.

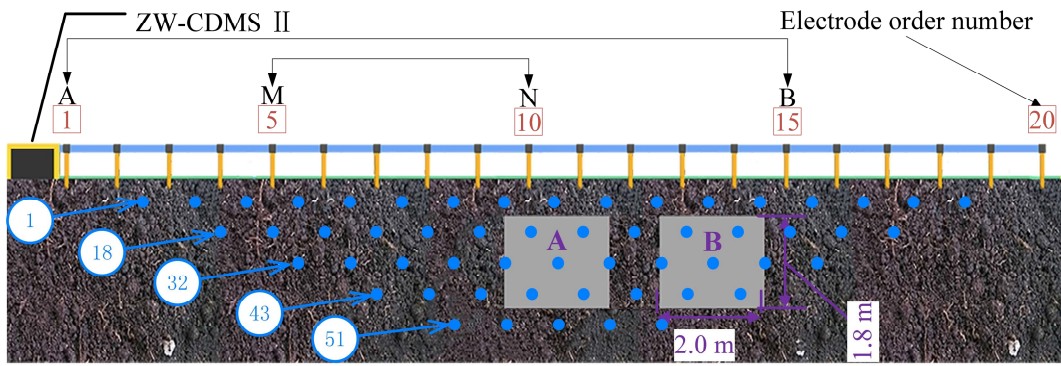

**Figure 11.** The schematic of the two high resistance cavities.

Figure 12 demonstrates the experimental SSIP data processed by the three algorithms, inverted with Res2DInv (Arifin et al., 2019). It can be observed that the location and shape of two abnormal bodies are distinguished only in the ECI algorithm while recognized as one whole in the other algorithms. We believe the reason that ECI has higher detection precision is due to its higher denoising ability.

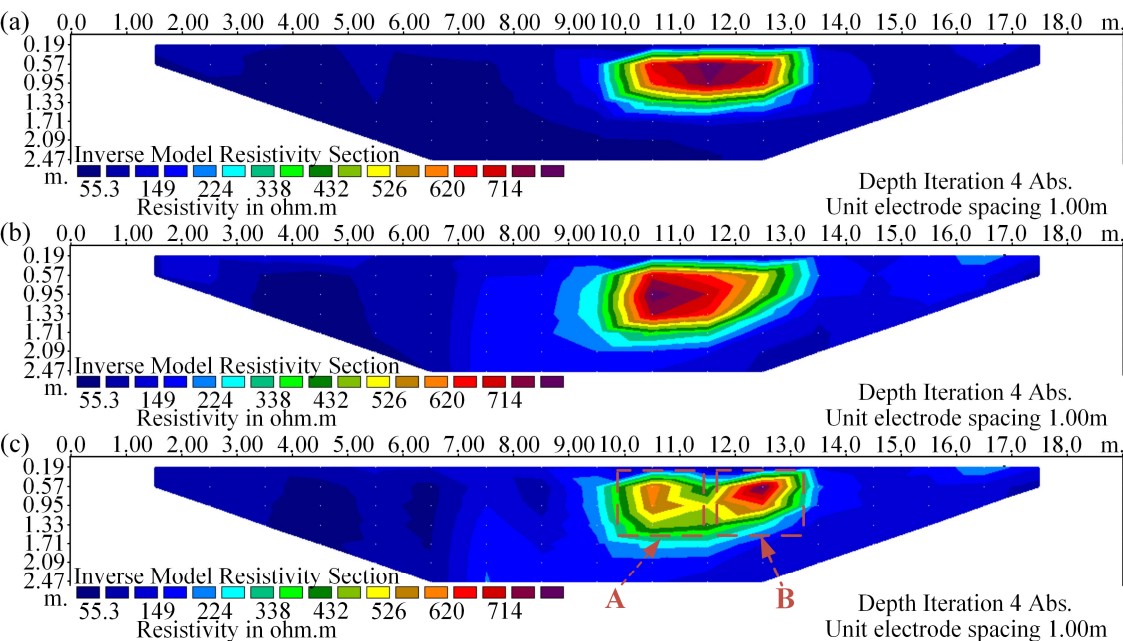

**Figure 12.** Inverted resistivity sections of the two high resistivity anomalies (A and B) at 80Hz with using **(a)** the FDIP method, **(b)** the TSIP algorithm, and **(c)** the ECI algorithm.

To verify the reason for the improved detecting precision, the SDs of data points are calculated from 18 to 50 (Figure 11), as shown in Figure 13. This figure shows that among the 33 SD of SSIP data processed by the ECI method is the lowest at all points. The average SD in ECI processing the SSIP data is 7% and 3.8% lower than the FDIP and TSIP, respectively. Also, the maximum value of SDs with the ECI method is 5% and 1.4% lower than the others, and the minimum value is 8% and 10% lower, respectively.

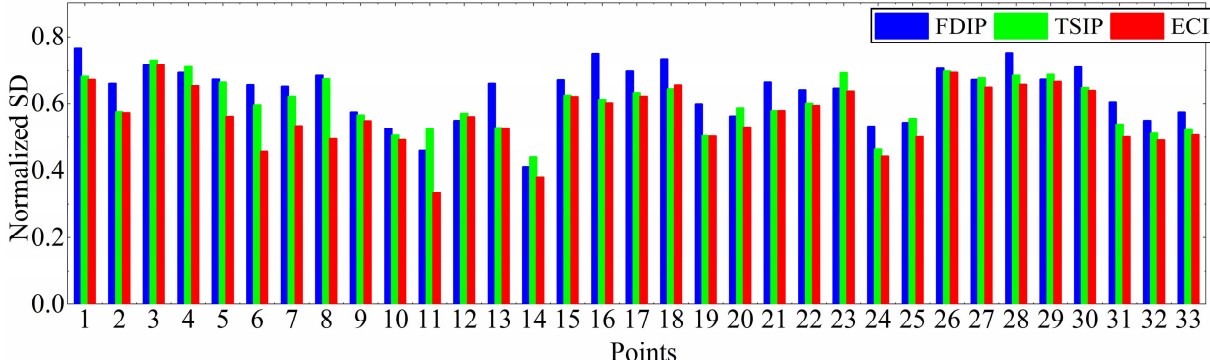

**Figure 13.** Standard deviation (SD) of the ECI algorithm and the others to the data dots from 18 to 50 at 80Hz.

Meanwhile, amplitude-frequency $|\rho(f)|$ and phase-frequency $\varphi(f)$ characteristics of complex resistivity are calculated by the three algorithms (one period) in survey point No 38, in Figure 11.

For example, Figures 14a1 and 14a2 show that the amplitude and phase of complex resistivity spectrum for this point at 80 Hz processed by FDIP are 39.7 Ω•m and -0.0881 rad, the amplitude and phase are 40.9 Ω•m and 6.12 rad when at 160Hz, and the amplitude and phase are 38.7 Ω•m and -0.253 rad when at 320Hz. As depicted in Figure 14, the complex resistivity processed by the ECI shows a linear trend with the three main frequencies. Also, the SD of the amplitude-frequency $|\rho(f)|$ characteristic is 0.10 and 0.49 lower than the others, and the SD of the phase-frequency $\varphi(f)$ is 3.56 and 0.03 lower. Therefore, we believe that the ECI algorithm has an advantage in suppressing background noise, which benefits the subsequent steps in SSIP data processing and imaging.

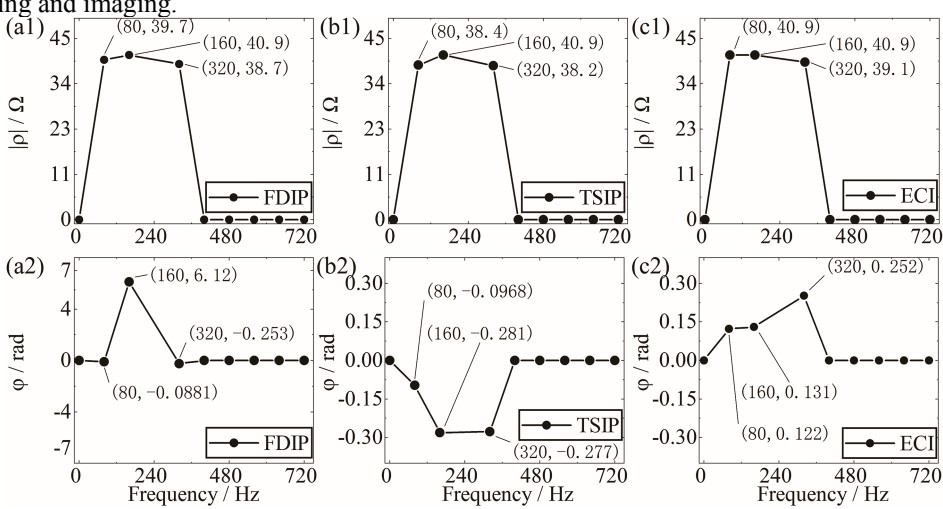

**Figure 14.** Complex resistivity spectrum calculated by the three algorithms (one period) in survey point No 38.

## 5 Discussion

The simulation results indicate that the ECI algorithm has very good performance in noise reduction and robustness. Along with the increase of the Gaussian noise level, we found that the ECI algorithm can to some extend overcome the shortcoming that the TSIP algorithm is susceptible to the noise of the current. This result coincided with Eq. (6) and Eq. (13), which provides a novel approach for correlated identification noise reduction. In the impulsive noise experiment, we found that the ECI algorithm still has

good noise reduction when the discrete point is more than 60%, which compensates for the disadvantage of the traditional denoising algorithm. Moreover, these simulation results also reveal that the ECI algorithm should have high robustness.

The standard deviations analysis of the real data indicates that that the ECI algorithm improves the accuracy and robustness of the collected data, which are compatible with the simulation analyses. This consistency shows that the ECI algorithm can obtain the location and shape of two abnormal bodies by improving the SNR of SSIP data, which can increase the resolution of inversion results.

## 6 Conclusions

We propose the ECI algorithm that effectively attenuates the background noise in SSIP data and improves the complex resistivity spectrum. This algorithm uses the correlation function to neutralize the influence of the background noise in the SSIP data, and the spectrum complex resistivity can be calculated at multiple frequencies by the formula of the complex resistivity. Simulation results show that the ECI algorithm can effectively attenuate the background noise and improve the SNR. Subsequently, the practicability of the ECI algorithm is further verified by a field test. The results demonstrate that the SD of the SSIP data is improved, which benefits the accuracy and stability of the collected data. There is a good agreement between the complex resistivity and the geological target body with high resistance, which indicates that the ECI algorithm can help to improve the quality of interpretation and inversion in the survey area. For the amplitude spectrum, the ECI algorithm can more effectively suppress the background noise, including the Gaussian random and impulsive noises. Still, its effect is very limited for the phase spectrum. Therefore, denoising algorithm based on pseudo-random sequence correlation identification is still left open for more investigation.

*Code availability.* The code is a collection of routine in MATLAB (MathWorks) and is available upon request to the author (e-mail: hsmfly1982@163.com).

*Data availability.* ALL the SSIP data are collected by ZW-CMDSII and are available upon request to the author (e-mail: hsmfly1982@163.com).

*Author contributions.* SH and YW designed the study, performed the research, analyzed data, and wrote the paper. JG contributed to language polishing and response. XJ and HX contributed to refining the ideas, carrying out additional analyses, and finalizing this paper.

*Competing interests.* The authors declare that they have no conflict of interest.

*Acknowledgements.* We are grateful for the help of J. Wang, S. Zhu, H. Wang and J. S. Cui. We thank the editors and the reviewers for the constructive comments that helped to improve this article.

*Financial support.* This research has been supported by Key technology projects of science and Technology Department of Jilin Province Scientific (20190303015SF), Research Project of Jilin Provincial Department of Education (JJKH20210692KJ and JJKH20211053KJ) and the Fundamental Research and Theme Funds for Changchun Institute of Technology, China.

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
