# Peer review of "An enhanced correlation identification algorithm and its application on spread spectrum induced polarization data"

_Nonlinear Processes in Geophysics, 2020_

## Referee Comment (RC1) · Anonymous Referee #1 · 14 Sep 2020

The manuscript titled "An enhanced correlation identification algorithm and its application on spread spectrum induced polarization data" by authors He et al discusses a new statistical method for noise suppression of electrical IP data, captured in the time-domain, but analyzed in the frequency domain.

While I'm familiar with spectral induced polarization methods, I have no first-hand experience with the discussed spread-spectrum induced polarization methods. Therefore my comments should be interpreted as either real issues to be dealt with, or (hopefully the case) just misunderstandings on my side (for which I apologize). In any case, I think a thorough rework of the _representation_ of the manuscript can remediate all

issues.

My major concern with the manuscript is the total lack of multi-frequency complex data, be it in the form of resistivity magnitude and phase values, or real and imaginary parts (spectra). My understanding of the method is that this is exactly the goal here: Capture multi-frequency complex impedance data using very fast time-domain-alike injection/measurement schemes. As such I strongly suggest to:

1) show extracted measured spectra (i.e., the resulting "data" after noise removal and conversion to the frequency domain): |R(f)|, Phase(f)

2) show phase results for the inversion results (if not easily included into the main text, additional magnitude/phase results for different frequencies could be supplied in a supplement)

From an outside view this is important to actually judge the method and the achievements presented in this study.

Other comments:

* Please employ a proper notation for complex entities. From my point of view most equations contain complex values.

* page 4,line 17: "ZW-CMDSII" not defined - please provide a reference here as this seems to be related to the construction of a measurement system?

* page 4/ECI approach: My understanding of the main point of the ECI approach is that you assume uncorrelated noise for applied voltage (n1), measured voltage (n2), and measured injected current (n3) (otherwise the cross-correlations between those quantities shouldn't be zero, as stated in page 4, line 19). If this reading of mine is correct, I would like to see more discussion on this: Is this always the case? What about electronic noise (e.g., from the ADCs involved) - shouldn't this superimpose on all three noise components? I understand that this is probably not an issue here, but it would be nice if you could guide the reader into the right direction. Also, this main

assumption should be more prominently presented in your manuscript.

* Is NT specified? What interval was used for the synthetic and the experimental cases?

* Fig 4: i(t) is the injected current, if I'm not mistaken. As such it should have a unit of Ampere, not Volts (same with the corresponding noise n3). If i(t) somehow has units of volts, please explain correspondingly (also in the previous text passages).

* Synthetic case: It is not clear to me why you only add noise to the injected current and not to the measured voltage (n2) and the applied voltage (n1). Isn't that the whole point of your study? It would be nice if you could show that also for the synthetic case the cross-correlation of those noise components reduces to zero (p4, line19). I suppose this entails generating suitably uncorrelated random ensembles.

While I think this step was taken to simplify the discussion, I think the simplification does not represent the problem at hand. As you stated in page 4, you also expect significant noise levels in n2 ("n2(t) and n3(t) may possess more massive energy..."), you should at least add suitable noise levels to n2 to test you algorithm. I still wonder why the current measurement entails such large noise components, given that this measurement is usually just a voltage measurement over a shunt resistor...

* Eq. 14/15: Is M defined in the text?

* Inversion results (Fig. 9: You do not discuss any error parameterization. However, I think this is crucial here in order to properly judge and understand the results: Did you account for the remaining noise components in each of the three subplots differently, or did you assume similar data noise estimates for the inversions? What were the final RMS values?

* Please use the same colorbar limits for all plots in Fig. 9. Otherwise a proper comparison is not possible.

* It would be nice if you could conduct a residual analysis of the inversions, comparing

the forward response of the final inversion model to the actual data. Does this analysis also follow the observed noise levels?

* Just to be sure (and perhaps encourage a slight extension of the last paragraph of the introduction to better clarify): The major point of this manuscript is that it takes into account also noise from the current measurement, which is not commonly done, right? For example Liu et al 2017, (10.1190/GEO2016-0109.1) seem to only assume noise on the primary potential measurements (The geophysics-paper also nicely shows pseudosections of both magnitude and phase - this would also be interesting here).

In conclusion, I suggest to improve the presentation of the manuscript and to better work out the novel contribution of the ECI algorithm in comparison to the various other correlation-based noise-reduction algorithms out there, as well as to make sure your test cases compare to those of other studies (i.e., current and voltage noise).

Looking forward to reading the published paper!

Best regards

---

## Referee Comment (RC2) · Anonymous Referee #2 · 28 Sep 2020

The authors propose an algorithm to attenuate the background noise in Spread Spectrum Induced Polarization data to improve complex resistivity spectra.The simulation results show that the algorithm can effectively attenuate the background noise and improve the signal to noise ratio. Real data obtained in a field test show tthat the algorithm reduces the standard deviation of the collected data.

The work is definetely valuable to IP practioners, and it should be published after a major revision.

1 - My general comment in that the results presented in manuscript are not discussed. Additionally, figure captions need to expand and explain the figure in a brief and simple

way, so the reader doesn't need to switch back to the tex to understand the figure.

2 - I have read the first review of the manuscript, and I agree with the reviewer that it's essencial to show multi-frequency and phase data.

3 - Page 7: "ECI algorithm still has superior denoising performance and holds smaller volatility of the relative error when the percentage of the outliers is more significant than 50% ." What do you mean by volatility here? This sentence is unclear.

4 - What situations would the algortim fail? That is, what are the limitations? Please show the limitations in detail (simulation or measured data and discussion).

5 - Phase (or quadradture component) results must be presented and discussed, otherwise we are not looking at the IP effects.

The work has potencial for a great paper, but it needs to show the results in more detail (phase or quadrature terms) and discuss them.

Please take my critiscim as a way to improve your manuscript, which I am looking forward to seeing published.

---

## Author Comment (AC1) · 26 Nov 2020

26-Nov-2020 Re: comments of Reviewer 1

Dear Reviewer 1:

We would like to express our sincere gratitude to you for your time and effort in handling our manuscript (npg-2020-8) entitled "An enhanced correlation identification algorithm and its application on spread spectrum induced polarization data", as well as providing us many constructive comments for improving very much the presentation and quality of this manuscript. It is worth pointing out that your comments and suggestions

have really constructively helped us improve further the quality and presentation of the manuscript. In light of their inspiring comments and suggestions, we have revised the manuscript duly and carefully, and the specific responses to the reviewers are listed in the supplement.

Sincerely, Dr. Siming He hsmfly1982@163.com

Please also note the supplement to this comment:
https://npg.copernicus.org/preprints/npg-2020-8/npg-2020-8-AC1-supplement.pdf

---

## Author Response (AR1)

26-Nov-2020

Re: comments of Reviewers 1 and 3

Dear Ms Natascha Töpfer:

We would like to express our sincere gratitude to the editors and anonymous reviews for their time and effort in handling our manuscript **(npg-2020-8)** entitled **"An enhanced correlation identification algorithm and its application on spread spectrum induced polarization data".**

We would like to say thanks again sincerely to the editors and anonymous reviews for their time and effort spent in handing our paper, as well as providing us many constructive comments for improving very much the presentation and quality of this manuscript.

It is worth pointing out that the reviewers' comments and suggestions have really constructively helped us improve further the quality and presentation of the manuscript. In light of their inspiring comments and suggestions, we have revised the manuscript duly and carefully, and the specific responses to the reviewers are listed as below, with the corresponding revisions **highlighted in blue color** in the revised manuscript.

Sincerely,
Dr. Siming He
hsmfly1982@163.com

We greatly appreciate your suggestions, and we hope our revisions have addressed your questions and made this manuscript better.

**Comment 1.** 1.show extracted measured spectra (i.e., the resulting "data" after noise removal and conversion to the frequency domain): |R(f)|, Phase(f).

**Response**.

We added Phase(f) to the experiment on synthetic SSIP data record. Figures 1 and 2 show the relative error of Phase(f) are calculated and compared at the three main frequencies when the noise RMS ranges from 0 to 0.9.

[Figure]

Figure 1.    The effect of different degrees of Gaussian noise to the measures excitation signals in the phase-frequency characteristics. (a) SNR of the polluted potential signal. Complex resistivity relative error at (b) 80 Hz, (c) 160 Hz, (d) 320 Hz comparison using the three methods.

[Figure]

Figure 2. The effect of different levels of spike noises to the measured excitation signals. (a) SNR of the contaminated potential signal in the phase-frequency characteristics. Complex resistivity relative error at (b) 80 Hz, (c) 160 Hz, (d) 320 Hz comparison using the three methods.

From Figures 1 and 2, the results do not reflect the noise reduction performances of the three algorithms. Therefore, these results are not put into our manuscript. But |R(f)| and Phase(f) processed by three algorithms reflect their noise reduction performance well in the field experiment, as shown in Figure 3. So |R(f)| and Phase(f) in the field experiment are added to our manuscript.

This information is added on Page 9, line 16 and 17, Page 10, line 1 to 5 and Figure 3.

[Figure]

Figure 3. Complex resistivity spectrum calculated by the three algorithm (one period) in survey point No 21.

**Comment 2.** show phase results for the inversion results (if not easily included into the main text, additional magnitude/phase results for different frequencies could be supplied in a supplement)

**Response.**

The phase results for the inversion results given by Res2DInv software is shown as Figure 4. As we can see, this figure shows chaos information from the phase shift between $i(t)$ and $u(t)$ of the

three methods. This is because the loss layer surrounding the shelter is very weak polarized, so the phase shift is very small, which cause the result easily contaminated by some random factors. Therefore, we have not figured out how to extract meaningful information out of the phase results, but it is possible that with proper strong-polarized experiment field there could be different results. We are eager to explore that in our next experiment plan.

[Figure]

Figure 4. Inverted phase sections of the two high resistivity anomalies at 80Hz with using (a) the FDIP method, (b) the TSIP algorithm, and (c) the ECI algorithm.

**Comment 3.** Please employ a proper notation for complex entities. From my point of view most equations contain complex values.
**Response.**
We have modified some complex entities.
Page 4, line 19 to 23

**Comment 4.** page 4,line 17: "ZW-CMDSII" not defined - please provide a reference here as this seems to be related to the construction of a measurement system?
**Response.**
We have added reference the reference of ZW-CMDSII (Zhang, et al., 2014; He, et al., 2014;) to the paper.
Page 6, line 5 to 12

References
He, G. Wang, J. Zhang, B. Y. Li, M. and Ma, C.: Design of High-density Electrical Method Data Acquisition System, Instrument Technique and Sensor, 2014, 8, 18-19, https:// doi.org/10.3969/j.issn.1002-1841.2014.08.007.
Zhang, B. Y. He, G. and Wang J.: New High-density Electrical Instrument Measuring System. Instrument Technique and Sensor, 2014, 1, https://doi.org /10.3969/j.issn.1002-1841.2014.01.009.

**Comment 5.** page 4/ECI approach: My understanding of the main point of the ECI approach is that you assume uncorrelated noise for applied voltage (n1), measured voltage (n2), and measured injected current (n3) (otherwise the cross-correlations between those quantities shouldn't be zero, as stated in page 4, line 19). If this reading of mine is correct, I would like to see more discussion on this: Is this always the case? What about electronic noise (e.g., from the ADCs involved) - shouldn't this superimpose on all three noise components? I understand that this is probably not an issue here, but it would be nice if you could guide the reader into the right direction. Also, this main assumption should be more prominently presented in your manuscript.

**Response.**

What you understand is basically right. In our experiment, the applied voltage $u_T(t)$ is measured within the powering system, so the environment interference only introduce noise into the measured voltage and measured infected current. In fact, $n_1(t)$ mainly consists of the floor noise of the measuring instruments (including ADCs noise), and very feebly influenced the coupling effect of $n_2(t)$ and $n_3(t)$, while $n_2(t)$ and $n_3(t)$ mainly consist of the environment noise. This is why we consider the cross-correlation between $n_1(t)$ $n_2(t)$ or $n_1(t)$ $n_3(t)$ 'approximately' zero. We adjust the expressions in page 4 line 13 to 15 to make it more clear. As for the electronic noise, since they are Gaussian noise, Figure 5 shows how its energy is compressed by the cross-correlation computation. But what you suggest inspires us to more thoroughly consider different noise sources. So we make further discussions on the correlation behaviors of these noise as below.

In a real environment, this model is contaminated by the environment interference and measuring instrument. It can be categorized into three types: the Gaussian random noise, the impulse interference, and the particular frequency disturbance (Wang and Li, 1986; Yan et al., 2016).

For our system, we assume the three noises are linearly overlapping on the three sensors, along with some weak influence of coupling effects. So, the noises in the three sensors are only different in amplitude. Hence,

$$n_1(t) = B_1 g(t) + C_1 p(t) + D_1 s(t) \tag{1}$$

$$n_2(t) = B_2 g(t) + C_2 p(t) + D_2 s(t) \tag{2}$$

$$n_3(t) = B_3 g(t) + C_3 p(t) + D_3 s(t) \tag{3}$$

where $n_k(t)$ is the noise in sensor $Y_k$, $k = 1,2,3$, respectively. $g(t)$, $p(t)$ and $s(t)$ are

separately Gaussian random noise, impulsive noise and particular frequency interference. $B_k, C_k$

and $D_k$ are the amplitudes of $g(t)$, $p(t)$ and $s(t)$, $k = 1,2,3$, respectively.

According to the properties of the correlation function, the cross-correlation results of the three kind of noise is as below:

A. For the Gaussian random noise, when $-NT \leq \tau \leq NT$ and $\tau \neq 0$, $R_{gg}(\tau)$ is shown in Figure

5.

[Figure]

Figure 5. Waveform and autocorrelation for the Gaussian random noise $g(t)$. (a) its time domain waveform.

(b) its autocorrelation $R_{gg}(\tau)$.

B. For the impulsive noise, when $-NT \leq \tau \leq NT$ and $\tau \neq 0$, it is considered that $R_{pp}(0) \gg R_{pp}(\tau)$, as shown in Figure 6.

[Figure]

Figure 6. Waveform and autocorrelation for the impulsive noise $p(t)$. (a) its time domain waveform

containing 20% of the outliers. (b) its autocorrelation $R_{pp}(\tau)$.

C. For the particular frequency disturbance, its autocorrelation has the same frequency with it, but when it is less effective for the transmitter output signal $u_{ab}(t)$ than that of the $u_{mn}(t)$ and

$i(t)$, $D_1 D_2 R_{ss}(\tau)$ and $D_1 D_3 R_{ss}(\tau)$ can be effectively suppressed, as shown in Figure 7.

[Figure]

Figure 7. Waveform and autocorrelation for the particular frequency interference $s(t)$. The power-line interference (a) at $D_1 = 0.01$, (b) at $D_2 = 1$. (c) their cross-correlation $D_1 D_2 R_{ss}(\tau)$.

Based on the analysis above, it can be concluded that the influences of Gaussian random and impulsive noises are more effectively suppressed, while the particular frequency disturbance is attenuated to some degree when the noise is in lower intensity. Therefore, the proposed method has more value on denoising for Gaussian and impulsive random noises.

(2) In our experiment design, the noise sources and the system are considered independent and linearly superpositioned. Since our method demonstrates enhanced denoising ability to both kinds of noise, we think it is reasonable to say it has better denoising method. To further verify this assumption, we conduct an experiment on the denoising of mixed noises, as shown in Figure 8.

[Figure]

Figure 8. The injected current contaminated by simulated by sum of synthetic different degrees of Gaussian random noise and synthetic different levels of impulsive random noises, and RECR compared at three frequencies.

(a) SNR of the contaminated injected current. RECR at (b) 80 Hz, (c) 160 Hz, (d) 320 Hz comparison using the hybrid method and the others in the amplitude-frequency characteristics.

References

Wang, F.S., and Li, T., 1986, Industry of stray current resistivity observation and avoid interference distance: Northeastern Seismological Research, 2, 44–48. doi:10.13693/j.cnki.cn21-1573.1986.02.006

Yan, T. J., Wang, S. Q., Mang, Y. X., and Luo, X. Z., 2016, Influence of human interference on application of electrical prospect-ing and corresponding anti–interference measures , Mineral Exploration, 7, 634–639. doi:10.3969/j.issn.1674-7801.2016.04.016

**Comment 6.** Is NT specified? What interval was used for the synthetic and the experimental cases?
**Response.**
(1)  NT is set 0.0125s in our experiment.
The information is added on Page 4, line 8 and Figure 4.
(2)  The time interval is 0.0016ms, with sampling frequency of 625kHz.
Page 8, line 12

**Comment 7.** Fig 4: i(t) is the injected current, if I'm not mistaken. As such it should have a unit of Ampere, not Volts (same with the corresponding noise n3). If i(t) somehow has units of volts, please explain correspondingly (also in the previous text passages).
**Response.**

Thank you for pointing out our negligence. The relationship of the injected current signal between electrode A and electrode B is:

$$i(t) = u_i(t)/R_s ,\qquad\qquad(1)$$

where, $R_s$ is a $1\Omega$ sampling resistor, and $u_i(t)$ is the voltage at the sampling resistor.

Figure 4 is modified and the introduction is added on Page 5, line 10 to 14.

**Comment 8.** Synthetic case: It is not clear to me why you only add noise to the injected current and not to the measured voltage (n2) and the applied voltage (n1). Isn't that the whole point of your study? It would be nice if you could show that also for the synthetic case the cross-correlation of those noise components reduces to zero (p4, line19). I suppose this entails generating suitably uncorrelated random ensembles.
**Response.**
In our experiment the measurement line is 19m and in a stable environment, so we consider the system linear time-invariant and the noise from the current and voltage measurement are linearly superpositioned (Pelton, et al., 1983; De, et al., 1983; Vinegar and Waxman, 1984; De, et al., 1992; Garrouch and Sharma, 1998). Therefore, it is actually equivalent whether the noise is added to the injected current $i(t)$, the measured potential signal $u(t)$ or the applied voltage $u_T(t)$, the

equivalent relationship is described as below

(1) When the measured potential signal $u(t)$ and the injected current $i(t)$ are contaminated by the noise $n_2(t)$ and the noise $n_3(t)$, we can obtain

$$y_1 = u_T(t) + n_1(t), \tag{1}$$

$$y_2 = u(t) + n_2(t), \tag{2}$$

$$y_3 = i(t) + n_3(t), \tag{3}$$

The cross-correlation functions can be expressed as follows:

$$R_{y_1y_2}(\tau) = R_{x_1x_2}(\tau) + R_{n_1n_2}(\tau), \tag{4}$$

$$R_{y_1y_3}(\tau) = R_{x_1x_3}(\tau) + R_{n_1n_3}(\tau), \tag{5}$$

Thus we can assume that the cross-relation between $n_1(t)$ and $n_2(t)$, $n_3(t)$ $R_{n_1n_2}(\tau) \approx 0$ and $R_{n_1n_3}(\tau) \approx 0$, we can further obtain:

$$R_{y_1y_2}(\tau) \approx R_{x_1x_2}(\tau) \tag{6}$$

$$R_{y_1y_3}(\tau) \approx R_{x_1x_3}(\tau) \tag{7}$$

(2) When the noise $n(t)$ from the current and voltage measurement are linearly superpositioned, $n(t) = n_1(t) + n_2(t)$ and the injected current $i(t)$ are contaminated by the noise $n(t)$, we can obtain

$$y_1 = u_T(t) + n_1(t), \tag{8}$$

$$y_2 = u(t), \tag{9}$$

$$y_3 = i(t) + n(t) = i(t) + n_2(t) + n_3(t), \tag{10}$$

The cross-correlation functions can be expressed as follows:

$$R_{y_1y_2}(\tau) = R_{x_1x_2}(\tau), \tag{11}$$

$$R_{y_1y_3}(\tau) = R_{x_1x_3}(\tau) + R_{n_1n_2}(\tau) + R_{n_1n_3}(\tau), \tag{12}$$

Thus we can assume that the cross-relation between $n_1(t)$ and $n_2(t)$, $n_3(t)$ $R_{n_1n_2}(\tau) \approx 0$ and $R_{n_1n_3}(\tau) \approx 0$, we can further obtain:

$$R_{y_1y_2}(\tau) \approx R_{x_1x_2}(\tau) \tag{13}$$

$$R_{y_1y_3}(\tau) \approx R_{x_1x_3}(\tau) \tag{14}$$

Based on the analysis above, it can be concluded that Eq. (6) and Eq. (7) are consistent with Eq. (13)

and Eq. (14).This is why we only add noise on the injected current to represent the overall noise summation.

The cross-correlation results are shown in Figures 5 and 6 in response 5

**Comment 9.** While I think this step was taken to simplify the discussion, I think the simplification does not represent the problem at hand. As you stated in page 4, you also expect significant noise levels in n2 ("n2(t) and n3(t) may possess more massive energy..."), you should at least add suitable noise levels to n2 to test you algorithm. I still wonder why the current measurement entails such large noise components, given that this measurement is usually just a voltage measurement over a shunt resistor...

**Response.**

Sorry we did not put this clear enough in the manuscript. What we actually did was adding noise on the injected current to represent the overall noise summation in the system, as we explained in Response 8.

Since we consider our system as a linear time invariant system, whether the noise is added on the drive signal $u_T(t)$, the measured potential $u(t)$ or the current $i(t)$ are equivalent. When observing the field measured data, we found that the current signal is more heavily interfered by noise, as shown in Figure 1, so we decided to add noise on the supply current.

[Figure]

Figure 9. The time waveform of measured potential and supply current on electrode No. 53

**Comment 10.** Eq. 14/15: Is M defined in the text?

**Response.**

M is the length in the text as follows:

Page 5, line 12

**Comment 11.** Please use the same colorbar limits for all plots in Fig. 9. Otherwise a proper comparison is not possible.

**Response.**

Yes, we use the same colorbar limits for all plots in Figure 10 as follows:

Page 9, line 5

[Figure]

Figure 10.    Inverted resistivity sections of the two high resistivity anomalies at 80Hz with using (a) the FDIP method, (b) the TSIP algorithm, and (c) the ECI algorithm.

**Comment 12.** the forward response of the final inversion model to the actual data. Does this analysis also follow the observed noise levels?

**Response.**

Sorry we are not so sure what you are referring to. In the actual field data, it is hard to extract the actual noise component. So we are trying to observe the noise level by calculating the SDs respectively, which help us evaluate the fluctuation degree of the processed signal, as shown in Figures 10 and 11.

As the figure shows, the ECI method has the lowest SD, which is why we deduce that our method has better denoising ability.

Page 9, and Page 10 line 1 to 6

**Comment 13.** Just to be sure (and perhaps encourage a slight extension of the last paragraph of the introduction to better clarify): The major point of this manuscript is that it takes into account also noise from the current measurement, which is not commonly done, right? For example Liu et al

2017, (10.1190/GEO2016-0109.1) seem to only assume noise on the primary potential measurements (The geophysics-paper also nicely shows pseudosections of both magnitude and phase - this would also be interesting here).

**Response.**

(1) In our field experiment, we did observe different noise levels both in potential and current measurements, in which the fluctuation in the current measurement is even acuter, as Figure. 11 shows.

[Figure]

Figure 11. The time waveform of measured potential and supply current on electrode No. 53

(2) As stated Page 3, line 13 and 14, and Page 4, line 4 and 5 in our manuscript, that the denoising ability of the TSIP algorithm is limited is caused by that $i(t)$ is sensitive to $n_3(t)$. To solve this problem, the ECI algorithm is proposed. Therefore, that $i(t)$ is added noise to verify the noise reduction performance of the ECI algorithm.

(3) In fact, we think the it should represent the total noise summation in the system whether the noise component is added on the potential or current during calculation. From our understanding, the difference between us and Liu et al. is that they put this component in 'acquired potential $U_0$' (Eq. (5)), and we put it in $I_{ab}$. As we were trying to explain in response 8, this to operation should be equivalent during the cross-correlation process.

To clarify this, we added some explanations in the experiment section.

Page 5, line 17 and 24

**Comment 14.** In conclusion, I suggest to improve the presentation of the manuscript and to better work out the novel contribution of the ECI algorithm in comparison to the various other correlation-based noise-reduction algorithms out there, as well as to make sure your test cases compare to those of other studies (i.e., current and voltage noise).

**Response.**

As Liu. et al. mentioned, correlation noise-reduction algorithms applied on SSIP data processing is rarely reported, now all reported experiments we can find are Liu's research and 'Time-Domain Spectral Induced Polarization Based on Pseudo-random Sequence' (Li et al. 2013), as the TCI referring to in our manuscript. Liu's research is more of a screening method to find suitable signal sequence, rather than data processing method. Therefore what we can do is comparing our method with Mei Li's method. To be more accurate, we change 'TCI' to 'TSIP' when referring to Mei Li's

research in the manuscript.

Page2, line 3 and 4

Page 6, line 2

Page 7, line 12

Page 9, line 6

**Comment 2.** I have read the first review of the manuscript, and I agree with the reviewer that it is essencial to show multi-frequency and phase data.

**Response**.

We added Phase(f) to the experiment on synthetic SSIP data record. Figures 1 and 2 show the relative error of Phase(f) are calculated and compared at the three main frequencies when the noise RMS ranges from 0 to 0.9.

[Figure]

Figure 1.    The effect of different degrees of Gaussian noise to the measures excitation signals in the phase-frequency characteristics. (a) SNR of the polluted potential signal. Complex resistivity relative error at (b) 80 Hz, (c) 160 Hz, (d) 320 Hz comparison using the three methods.

[Figure]

Figure 2.    The effect of different levels of spike noises to the measured excitation signals. (a) SNR of the contaminated potential signal in the phase-frequency characteristics. Complex resistivity relative error at (b) 80 Hz, (c) 160 Hz, (d) 320 Hz comparison using the three methods.

From Figures 1 and 2, the results do not reflect the noise reduction performances of the three algorithms. Therefore, these results are not put into our manuscript. But |R(f)| and Phase(f) processed by three algorithms reflect their noise reduction performance well in the field experiment, as shown in Figure 3. So |R(f)| and Phase(f) in the field experiment are added to our manuscript.

This information is added on Page 9, line 16 and 17, Page 10, line 1 to 5 and Figure 3.

[Figure]

Figure 3.    Complex resistivity spectrum calculated by the three algorithm (one period) in survey point No 21.

**Comment 3.** Page 7: "ECI algorithm still has superior denoising performance and holds smaller

volatility of the relative error when the percentage of the outliers is more significant than 50% .”
What do you mean by volatility here? This sentence is unclear.

**Response**.

By 'volatility' we were trying to say 'fluctuation', sorry for causing misunderstanding.

Page 6, line 8

**Comment 4.** What situations would the algortim fail? That is, what are the limitations? Please show the limitations in detail (simulation or measured data and discussion).

**Response**.

In a real environment, this model is contaminated by the environment interference and measuring instrument. It can be categorized into three types: the Gaussian random noise, the impulse interference, and the particular frequency disturbance (Wang and Li, 1986; Yan et al., 2016).

For our system, we assume the three noises are linearly overlapping on the three sensors, along with some weak influence of coupling effects. So, the noises in the three sensors are only different in amplitude. Hence,

$$n_1(t) = B_1 g(t) + C_1 p(t) + D_1 s(t) \tag{1}$$

$$n_2(t) = B_2 g(t) + C_2 p(t) + D_2 s(t) \tag{2}$$

$$n_3(t) = B_3 g(t) + C_3 p(t) + D_3 s(t) \tag{3}$$

where $n_k(t)$ is the noise in sensor $Y_k$, $k = 1, 2, 3$, respectively. $g(t)$, $p(t)$ and $s(t)$ are separately Gaussian random noise, impulsive noise and particular frequency interference. $B_k, C_k$ and $D_k$ are the amplitudes of $g(t)$, $p(t)$ and $s(t)$, $k = 1, 2, 3$, respectively.

According to the properties of the correlation function, the cross-correlation results of the three kind of noise is as below:

A. For the Gaussian random noise, when $-NT \le \tau \le NT$ and $\tau \ne 0$, $R_{gg}(\tau)$ is shown in Figure 4.

[Figure]

Figure 4. Waveform and autocorrelation for the Gaussian random noise $g(t)$. (a) its time domain waveform.

(b) its autocorrelation $R_{gg}(\tau)$.

B. For the impulsive noise, when $-NT \leq \tau \leq NT$ and $\tau \neq 0$, it is considered that $R_{pp}(0) \gg R_{pp}(\tau)$,

as shown in Figure 5.

[Figure]

Figure 5. Waveform and autocorrelation for the impulsive noise $p(t)$. (a) its time domain waveform

containing 20% of the outliers. (b) its autocorrelation $R_{pp}(\tau)$.

C. For the particular frequency disturbance, its autocorrelation has the same frequency with it, but

when it is less effective for the transmitter output signal $u_{ab}(t)$ than that of the $u_{mn}(t)$ and $i(t)$,

$D_1 D_2 R_{ss}(\tau)$ and $D_1 D_3 R_{ss}(\tau)$ can be effectively suppressed, as shown in Figure 6.

[Figure]

Figure 6. Waveform and autocorrelation for the particular frequency interference $s(t)$. The power-line interference (a) at $D_1 = 0.01$, (b) at $D_2 = 1$. (c) their cross-correlation $D_1 D_2 R_{ss}(\tau)$.

Based on the analysis above, it can be concluded that the influences of Gaussian random and impulsive noises are more effectively suppressed, while the particular frequency disturbance is attenuated to some degree when the noise is in lower intensity. Therefore, the proposed method has more value on denoising for Gaussian and impulsive random noises.

Page 10, line 15 and 16

[Figure]

Figure 7.    Inverted phase sections of the two high resistivity anomalies at 80Hz with using (a) the FDIP method,
(b) the TSIP algorithm, and (c) the ECI algorithm.

(2) We added phase results to field experiment and not to simulation as we explained in Response
2.

[revised manuscript text omitted]

---

## Author Response (AR3)

20-Mar-2020

Re: comments of the editors and Reviewers 2

Dear Dr. Richard Gloaguen:

We would like to express our sincere gratitude to the editors and anonymous reviews for their time and effort in handling our manuscript **(npg-2020-8)** entitled **"An enhanced correlation identification algorithm and its application on spread spectrum induced polarization data".**
We would like to say thanks again sincerely to the editors and anonymous reviews for their time and effort spent in handing our paper, as well as providing us many constructive comments for improving very much the presentation and quality of this manuscript.
It is worth pointing out that the reviewers' comments and suggestions have really constructively helped us improve further the quality and presentation of the manuscript. In light of their inspiring comments and suggestions, we have revised the manuscript duly and carefully, and the specific responses to the reviewers are listed as below, with the corresponding revisions **highlighted in blue color** in the revised manuscript.

Sincerely,
Dr. Siming He
hsmfly1982@163.com

**Responses to comments of Reviewer 2**

**We greatly appreciate your suggestions, and we hope our revisions have addressed your questions and made this manuscript better.**

**Comment 1.** The reference list needs to be updated. It should be alphabetic and the year should be at the end. See https://www.nonlinear-processes-in-geophysics.net/submission.html#manuscript composition for guide lines.

**Response.**

We have fixed this problem, checked and modified the literature formatting carefully according to the formatting guide.

**Comment 2.** Høyer et al. 2018 is missing from the reference list.

**Response.**

We have added it to the reference list.

Page 12, Line 33 to 35

The related reference is as follows:

Høyer, A. S., Klint, K. E. S., Fiandaca, G., Maurya, P. K., Christiansen, A. V., Balbarini, N., Bjerg, P. L., Hansen, T. B., and Møller, I.: Development of a high-resolution 3D geological model for landfill leachate risk assessment. Engineering Geology, 249, 45–59, https://doi:10.1016/j.enggeo.2018.12.015, 2015.

**Comment 3.** Page 1: "SSIP technology has a certain degree of noise immunity".

**Response.**

Sorry, what we are trying to say is that one of the advantages of this sequence is to be essentially spectrally flat in a given frequency range, which can be used in noise reduction technology (Liu et al., 2017).

We have replaced "SSIP technology has a certain degree of noise immunity" to "In field detection experiments, it is still a major problem that IP data is often contaminated with background noise."

Page 1, Line 34

The related reference is as follows:

Liu, W. Q., Chen, R. J., Cai, H. Z., Luo, W. B., and Revil, André.: Correlation analysis for spread spectrum induced polarization signal processing in electromagnetically noisy environments, Geophysics, 82, E243–E256, https://doi.org/10.1190/geo2016-0109.1, 2017.

**Comment 4.** Page 9, Line 1: Reference missing for Res2DInv.

**Response.**

We have added its to the reference list.

Page 10, Line 3 and 4

Page 12, Line 8 to 10

The related reference is as follows:

Arifin, M. H., Kayode, J. S., Izwan, M. K., Zaid, H. A. H., and Hussin, H.: Data for the potential gold mineralization mapping with the applications of Electrical Resistivity Imaging and Induced Polarization geophysical surveys, Data in Brief, 22, 830–835. doi:10.1016/j.dib.2018.12.086, 2019.

**Comment 5.** Page 9, Line 8, "red point". I guess you are referring to Figure 8? Maybe just "data points"?

**Response.**

Thank you for pointing out our negligence. Data points are numbered according to Figure 11, and the related data points are modified.

Page 10 Line 1, 2 and 10

Page 11 Line 2

These modifications are as follows:

(1) To verify the reason of the improved detecting precision, the SDs of data points are calculated from 18 to 50 (Figure 10), as shown in Figure 11.

(2) Figure 13. Standard deviation (SD) of the ECI algorithm and the others to the data dots from No. 18 to 50 at 80Hz.

**Comment 6.** Page 9, Line 17, "algorithm" should be "algorithms"

**Response.**

We have replaced "algorithm" with "algorithms".

Page 11, Line 4

**Comment 7.** Figure 8: What is the difference between blue and red dots? If there is no difference, I suggest that you use the same color.

**Response.**

Many thanks for your suggestion. We have changed all the red dots to blue dots.

Page 10, Line 1 and 2

This modifications is as follows:

[Figure]

Figure 11. The schematic of the two high resistance cavities.

**Comment 8.** Figure 11: Please elaborate the figure text. What are the number in the plot?

**Response.**

We add more specific explanations for Figure 14 in Page 11, Line 5 to 7. This modification is as follows:

"For example, Figure 1(a) and (d) shows that the amplitude and phase of complex resistivity spectrum for this point at 80 Hz processed by FDIP are 39.7 $\Omega{\cdot}m$ and -0.0881 rad, the amplitude and phase are 40.9 $\Omega{\cdot}m$ and 6.12 rad when at 160Hz, and the amplitude and phase are 38.7 $\Omega{\cdot}m$ and -0.253 rad when at 320Hz."

**Comment 9.** Experiment:

**a. What is the resistivity of the two cavities. It is difficult to access the inversion results, when the resistivity is unknown.**

**Response.**

We employ the high-density resistivity method. This method is used to infer geological structure by utilizing the differences in the electrical conductivity between the loess and the two cavities. In the case of the known geological structure,we can verify the correctness of the inversion result according to the differences, as others have reported in the literature (Liu et al., 2017).

The related reference is as follows:

Liu, W. Q., Chen, R. J., Cai, H. Z., Luo, W. B., and Revil, André.: Correlation analysis for spread spectrum induced polarization signal processing in electromagnetically noisy environments, Geophysics, 82, E243–E256, https://doi.org/10.1190/geo2016-0109.1, 2017.

**b. As commented by previous review, you need to show some data and data fits. So, a figure with the recorded data (resistivity and phase) and the forward response (resistivity and phase) of the final inversion model. This could be added to Figure 11.**

**Response.**

We agree with your suggestion. Unfortunately, we did not have the forward response (resistivity and phase) about the model. As said in comment 9.a, the model is the known geological structure. The loess and the two cavities have the obvious conductive differences. Therefore, we can use the differences to infer the geological structure of the model. To verify the efficacy of the test system, we take a variety of measures.

(1) The loess is measured to have an electronic resistivity of $36\,\Omega$•m 。

(2) Diagram of the field-test is added to the paper.

(3) We replace the simulation experiment with the resistance-capacitance experiment, present the recorded data and the forward response,and analyze the noise reduction performance on the frequency-spectrum by the three algorithms.

Page 9, Line 9, and 10

Page 9, Line 12 and 13

Page 6, Line 8 and 9

This modifications is as follows:

(1) The two cavities are buried by loess. The loess is measured to have an electronic resistivity of 36 $\Omega$•m .

(2)

[Figure]

**Figure 10.** Diagram of the field-test.

(3)

[Figure]

**Figure 5.** (a) Experimental schematic; (b) Experimental setup.

**c. You try to sell the algorithm for handling IP data, so you need to show a section with the phase results. This was also pointed out by a previous reviewer. If you IP results are not satisfying due to low chargeability (as you write in your answer), then I suggest that you find another example. As a minimum, this should be included in a discussion.**

**Response.**

Yes, we replace the simulation experiment with the resistance-capacitance experiment, present the recorded data and the forward response,and analyze the noise reduction performance on the frequency-spectrum (amplitude and phase) by the three algorithms.

Page 5 to 8

This modifications are as follows:

To validate the effectiveness of the ECI system, we performed a resistance-capacitance experiment, as shown in Figure 5.

The circuit parameters are chosen to be $R_A = 30.3\Omega / 5W$, $R_{MN} = 30.1\Omega / 5W$ $R_B = 30\Omega / 5W$ and $C_{MN} = 470\mu F$. We recorded the applied voltage $u_T(t)$, the injected current $i(t)$ and the measured potential signal $u(t)$ as the raw signals. These signals are a 3-order spread spectrum pseudo-random sequence at the clock cycle of 0.0125s, as shown in Figures 6a-c and Table 1.

Since our experiment is in a stable environment, we consider the system linear time-invariant and the noise from the current and voltage measurement are linearly superpositioned (Pelton, et al., 1983; De, et al., 1983; Vinegar and Waxman, 1984; De, et al., 1992; Garrouch and Sharma, 1998). Therefore, it is actually equivalent whether the noise is added to the injected current $i(t)$, the measured potential signal $u(t)$ or the applied voltage $u_T(t)$. Therefore, the injected current $i(t)$ is only polluted by the synthetic background noise, including Gaussian and impulsive, as shown in Figures 6d and e. Thirdly, the complex resistivity of the main frequency is considered and discussed because the main energy of the pseudo-random signal is concentrated on the main frequency (He, 2017). Finally, for detailed comparisons between the ECI algorithm and the others, we add the synthetic Gaussian and impulsive noises to the response signal $i(t)$, respectively.

[Figure]

**Figure 5.**   (a) Experimental schematic; (b) Experimental setup.

[Figure]

**Figure 6.**   The time waves of (a) the applied voltage $u_T(t)$, (b) the measured potential signal u(t), (c) the voltage $u_i(t)$ at the sampling resistor

**Table 1.**   Amplitude and phase values of complex resistivity obtained with Figures 6a-c.

| Frequency (Hz) | Theoretical amplitude (Ω) | Theoretical phase (rad) | Measured amplitude (Ω) | Measured phase (rad) |
|---|---|---|---|---|
| 80.2 | 30.8 | -0.14 | 30.8 | -0.14 |
| 160.4 | 30.4 | -0.07 | 30.3 | -0.08 |
| 320.8 | 30.2 | -0.03 | 30.7 | -0.03 |

We use synthetic Gaussian noise with the deviation and mean values of 0.1 and 1.1 as a standard template. The excitation signal $i(t)$ is polluted by synthetic different energy levels of the Gaussian noise. Figure 7 show that the denoised results are obtained and compared at the three main frequencies when

the noise RMS ranges from 0.12 to 0.25. The figure shows that as the RMS of noise increases, the complex resistivity information obtained by each algorithm decreases. However, the amplitude spectrum after ECI processing has the slowest falling speed, and the phase spectrum has the slowest falling speed at 80 Hz.

[Figure]

**Figure 7.**    amplitude and phase of complex resistivity values at (a1) and (a2) 80 Hz, (b1) and (b2) 160 Hz, (c1) and (c2) 320 Hz comparison using the three methods.

Previous literature has shown that if the percentages of outliers in impulsive noise exceed $50\%$, the traditional denoising algorithm will be limited (Liu et al., 2016, 2017). Thus, Synthetic impulsive noise is added to the excitation signal $i(t)$ in ten percent steps. Their standard deviations (SDs) and skewnesses (SKs) are shown in Figure 8. As depicted in Figures 9, the three algorithms have a certain degree of denoising performance versus the different percentages of the synthetic outliers against the raw data. The figure shows that with the discrete points of impulse noise growing, the RMS of noise is different. The amplitude spectrum and phase spectrum of complex resistivity obtained by each algorithm fluctuate. Although the noise reduction performance of the phase spectrum processed by ECI does not stand out, the overall change of the amplitude spectrum after ECI processing is still slow, especially when the discrete point is more than 50%.

[Figure]

**Figure 8.**    The standard deviations (SDs) and skewnesses (SKs) of synthetic impulsive noise.

[Figure]

**Figure 9.** Complex resistivity values at (a1) and (a2) 80 Hz, (b1) and (b2) 160 Hz, (c1) and (c2) 320 Hz comparison using the three methods.

**Comment 10.** A discussion should be added in the manuscript with the answer to comment 5 from reviewer 1 (version 3).

**Response.**

Yes, we have added this discussion to the paper.

Page 4, Line 13 to 16.

This modifications is as follows:

Figure 4 shows the Schematic diagram of ZW-CMDSII (Zhang et al., 2014; He et al., 2014;). As is known from the figure, we are able to conclude that $u_\text{T}(t)$ is mainly disturbed by the floor noise energy of the instrument, and $i(t)$ and $u(t)$ are mainly contaminated by environmental noise. The floor noise is relatively very low, while environment noise possesses a much higher energy level. Thus we assume that $n_1(t) \approx 0$, and can conclude that zero correlation between $n_1(t)$ and $n_2(t)$, $n_3(t)$, $R_{m_1 n_2}(\tau) \approx 0$ and $R_{m_1 n_3}(\tau) \approx 0$.

[Figure]

**Figure 4.** Schematic diagram of the instrument.

**Comment 11.** Is the code available for other researchers to use? Please add a section with code availability after the conclusion.

**Response.**

Sorry that we cannot open the code for public due to the confidentiality agreements of the funding.

However, we can provide the algorithm code for interested researchers via email (hsmfly1982@163.com).

**Note:**

We also add an expression in Eq. 13 ($=\left|\dfrac{P_{y_1 y_2}(\omega)}{P_{y_1 y_3}(\omega)}\right| e^{-j(\varphi_{y_1 y_2}(\omega)-\varphi_{y_1 y_3}(\omega))}$) to better illustrate the relationship

between the amplitude and phase. This does not change any experiment results but we believe it could make the equations easier to understand.

Page 5, Line 5 and 6

---

## Author Response (AR4)

14-Apr-2020

Re: comments of the editors

Dear Dr. Richard Gloaguen:

We would like to express our sincere gratitude to the editors for their time and effort in handling our manuscript **(npg-2020-8)** entitled **"An enhanced correlation identification algorithm and its application on spread spectrum induced polarization data".**
We would like to say thanks again sincerely to the editors and anonymous reviews for their time and effort spent in handing our paper, as well as providing us many constructive comments for improving very much the presentation and quality of this manuscript.
It is worth pointing out that the reviewers' comments and suggestions have really constructively helped us improve further the quality and presentation of the manuscript. In light of their inspiring comments and suggestions, we have revised the manuscript duly and carefully, and the specific responses to the editors are listed as below, with the corresponding revisions **highlighted in blue color** in the revised manuscript.

Sincerely,
Dr. Siming He
hsmfly1982@163.com

**Responses to comments of Editors**

**We greatly appreciate your suggestions, and we hope our revisions have addressed your questions and made this manuscript better.**

**Comment 1.** I will have to recommend the rejection of this manuscript if a proper discussion is not included. The discussion should be clearly separated from the results and encompass the aspects raised by the reviewers and myself.

**Response.**

1)    Yes, we have added this discussion to the paper.

Page 10, Line 14 to 18.

Page 11, Line 1 to 6.

This modifications is as follows:

**5 Discussion**

The simulation results indicate that the ECI algorithm has very good performance in noise reduction and robustness. Along with the increase of the Gaussian noise level, we found that the ECI algorithm can to some extend overcome the shortcoming that the TSIP algorithm is susceptible to the noise of the current. This result coincided with Eq. (6) and Eq. (13), which provides a novel approach for correlated identification noise reduction. In the impulsive noise experiment, we found that the ECI algorithm still has good noise reduction when the discrete point is more than 60%, which compensates for the disadvantage of the traditional denoising algorithm. Moreover, these simulation results also reveal that the ECI algorithm should have high robustness.

The standard deviations analysis of the real data indicates that that the ECI algorithm improves the accuracy and robustness of the collected data, which are compatible with the simulation analyses. This consistency shows that the ECI algorithm can obtain the location and shape of two abnormal bodies by improving the SNR of SSIP data, which can increase the resolution of inversion results.

2)    Also, we have modified the conclusions.

Page 11, Line 15 to 18.

This modifications is as follows:

For the amplitude spectrum, the ECI algorithm can more effectively suppress the background noise, including the Gaussian random and impulsive noises. Still, its effect is very limited for the phase spectrum. Therefore, denoising algorithm based on pseudo-random sequence correlation identification is still left open for more investigation.

**Note:**

We also add the sections "Code availability", "Data availability", "Author contribution", "Competing interests", "Acknowledgements", and "Financial support".

Page 11, Line 20 to 37.

The sections is as follows:

*Code availability.* The code is a collection of routine in MATLAB (MathWorks) and is available upon request to the author (e-mail: hsmfly1982@163.com).

*Data availability.* ALL the SSIP data are collected by ZW-CMDSII and are available upon request to the author (e-mail: hsmfly1982@163.com).

*Author contributions.* SH and YW designed the study, performed the research, analyzed data, and wrote the paper. JG contributed to language polishing and response. XJ and HX contributed to refining the ideas, carrying out additional analyses, and finalizing this paper.

*Competing interests.* The authors declare that they have no conflict of interest.

*Acknowledgements.* We are grateful for the help of J. Wang, S. Zhu, H. Wang and J. S. Cui. We thank the editors and the reviewers for the constructive comments that helped to improve this article.

*Financial support.* This research has been supported by Key technology projects of science and Technology Department of Jilin Province Scientific (20190303015SF), Research Project of Jilin Provincial Department of Education (JJKH20210692KJ and JJKH20211053KJ) and the Fundamental Research and Theme Funds for Changchun Institute of Technology, China.